# Understanding for Prevention: Qualitative and Quantitative Analyses of Suicide Notes and Forensic Reports

**DOI:** 10.3390/ijerph20032281

**Published:** 2023-01-27

**Authors:** Yolanda Mejías-Martín, Celia Martí-García, Yolanda Rodríguez-Mejías, Ana Alejandra Esteban-Burgos, Víctor Cruz-García, María Paz García-Caro

**Affiliations:** 1Virgen de las Nieves University Hospital, 18014 Granada, Spain; 2Hygia Research Group, ibs.GRANADA, Health Research Institute, 18014 Granada, Spain; 3Nursing Department, Faculty of Health Sciences, University of Málaga, 29071 Málaga, Spain; 4Department of Nursing, Faculty of Health Sciences, University of Jaén, 23071 Jaén, Spain; 5Cinebase, Cinema and Audiovisual School of Catalonia (ESCAC), 08222 Terrassa, Spain; 6Nursing Department, Faculty of Health Sciences, University of Granada, 18071 Granada, Spain; 7Mind, Brain and Behavior Research Center (CIMCYC), University of Granada, 18071 Granada, Spain

**Keywords:** suicide, suicide prevention, suicide notes, forensic reports, qualitative research

## Abstract

Suicide risk is associated with vulnerabilities and specific life events. The study’s objective was to explore the relevance of data from forensic documentation on suicide deaths to the design of person-centered preventive strategies. Descriptive and thematic analyses were conducted of forensic observations of 286 deaths by suicide, including some with suicide notes. Key findings included the influence of health-and family-related adverse events, emotional states of loss and sadness, and failures of the health system to detect and act on signs of vulnerability, as confirmed by the suicide notes. Forensic documentation provides useful information to improve the targeting of preventive campaigns.

## 1. Introduction

Suicide is a global cross-cultural phenomenon considered responsible for around 800,000 deaths per year [1,2]. According to the latest prepandemic WHO Suicide worldwide report. In 2019, 1.3% of deaths were due to suicide, and the global age-standardized suicide rate was 9.0 per 100,000 inhabitants (2.3 times higher in men than in women). This potentially preventable act occurred before the age of 50 in over half of global suicides [3] and has a major emotional impact on loved ones [4]. The development of preventive strategies requires an understanding of why individuals attempt suicide, which may involve multiple psychological, social, biological, cultural, and environmental risk factors [2,4].

According to data presented in 2015 by the Statistical Office of the European Union (Eurostat) related to the number of deaths by suicide [5], Spain is ranked 23rd out of 28 countries, with a ratio of 7.51 deaths by suicide per 100,000 inhabitants. In Andalusia, they accounted for 4% of all potentially avoidable deaths in men and 2.8% in women in the period 2000–2019, for the population under 75 years of age [6].

Death by suicide is classified as a violent death in Spain [7,8], where any violent or suspicious death gives rise to a judicial process to establish its cause. This entails a medical–legal assessment and expert autopsy report [8], among other evidence, to determine the true dynamics of the event [9] and differentiate among homicides, suicides, and accident deaths [10,11]. When death by suicide is suspected, the forensic physician examines two aspects: (i) the cause, setting, and circumstances of the death and (ii) the motives, characteristics, and situation of the individual and any relevant signs [10,12]. These observations are included in the forensic autopsy report, which sometimes contains information from family members, friends, and/or neighbors on the individuals and their deaths. [10,11,13,14]. However, despite the importance of these data for developing well-targeted suicide prevention campaigns, there have been only limited attempts to improve the understanding of suicide through the study of forensic observations [9,13,15]. Therefore, the information is obtained in the first instance by direct observation of the scene of the events and by questioning the people present (relatives, neighbors, acquaintances). Later, the information obtained in the clinical autopsy process (concomitant causes, substance use, time of death, etc.) is added. This direct approach allows observation of where the person chose to die (in what area of the home, in the mountains, workplace, etc.) and in what way (method), how they prepared the scene (if it was an impulsive act or, on the contrary, it was carefully prepared, leaving even the necessary documentation near the body), and if the deceased left any note before he died.

Deaths by suicide often leave clues that allow interpretation of the event [9]. So-called suicide notes or letters can yield key information on the origin, time course, and severity of the distress that prompted the act [16,17,18]. They can reveal the degree of impulsivity and hopelessness, the life problems faced, and any signs of intentionality and planning [12]. Although not standard practice, examination of this information should be considered complementary to conventional forensic data, enhancing the understanding of the individual’s psychological state, especially in the moments leading up to a successful attempt [18,19,20]. However, this direct first-hand information is not available in the majority of cases, with the percentage of suicides leaving a note ranging widely between 3 and 45% in different studies, being typically around 25% [21,22,23,24]. Furthermore, as a data source, suicide notes have a series of limitations, such as small samples, a predominant focus on demographic constructs, and the absence of multivariate analysis [23].

Inadequate efforts to prevent suicide have been attributed to a lack of awareness of the problem and to prejudices that prevent its open discussion [4]. Suicide prevention is considered to require focused evidence-based interventions in an innovative, multi-sectoral, and comprehensive approach, involving not only health care but also education, police, and the legal system, among others [4,25,26,27,28,29]. In addition, there have been calls for a more person-centered approach based on the highly personal and fluid nature of suicide risk, which is related to individual vulnerabilities and person-specific events that trigger suicidal thoughts and actions [27,30,31]. This type of preventive strategy should greatly benefit from first-hand evidence of the thoughts that pass through the minds of individuals making this decision [17,21,22,32,33,34].

Neither suicide notes nor forensic reports about the conditions of death by suicide are available in standardized databases, and there is a need to conduct research at a more local level [13] for in-depth analysis of the social, cultural, and economic factors involved [35,36]. There has been little research in Spain on psychological autopsies, forensic observations, or suicide notes to explore influential factors [12,37,38]. Thus, forensic reports can offer personalized information when they include direct observations and information collected from family and friends about the conditions and circumstances of death. These person-centered observations may help and/or supplement the absence of suicide notes or psychological autopsy reports.

The objective of this study was to explore the circumstances and contexts of people dying by suicide in Granada (Southern Spain) between 2007 and 2013 through quantitative and qualitative analyses of forensic reports and suicide notes, exploring the relevance of findings to the design of preventive strategies.

## 2. Materials and Methods

### 2.1. Design

An observational, descriptive, retrospective study was undertaken, carrying out quantitative and qualitative analyses of forensic reports on people dying by suicide, including any suicide notes found.

### 2.2. Data Source

The study population was the inhabitants of the province of Granada from 2007 to 2013. This population ranged between 884,099 inhabitants in 2007 and 919,319 inhabitants in 2013 [6].

The study database was developed in a previous research project on suicide in Granada and contains data extracted from the records of the Forensic Anatomical Institute of Granada [39]. It includes information on the timing and method of suicides in the province and on the sociodemographic, clinical, pharmacological, personal/family, and other characteristics of people classified as dying by suicide in Granada.

Inclusion criteria were autopsy confirmed case of suicide between the 1 January 2007 and 31 December 2013 with the availability of a forensic report containing observations on the deceased and circumstances of the suicide, with or without reference to a suicide note. Data were gathered on the sex and age of individuals and on the month/year and method of suicide. Information was also collected from forensic reports and any suicide notes (literal transcription, summary, or other forensic references about the suicide note) on the background, condition, or situation of the individual prior to the act of suicide.

### 2.3. Participants

Forensic reports containing the previously described information were available for 286 (38.5%) of the 743 cases of death by suicide reported during the study period, which were therefore selected for this study (complementary information with the mandatory clinical autopsy data). The mean age was 52.99 years (±19.135), and the individual was male in 77.3% of these cases, and death was by hanging in 64%. The most frequent age group was 41–50 years (22.4%). Suicide notes were reported in 105 (36.7%) of the 286 cases, although only 18 (17.1%) were transcribed literally, while the remaining 87 were a summary or a reference to the content made by the forensic physician.

Forensic reports were most frequent in 2010 (20.6%) and least frequent in 2007 (8.4%); by month, they were most frequent in May (11.2%) and least frequent in November (5.2%) (Table 1).

In the subgroup of 105 cases with suicide notes, 77.1% were males and 66.6% were aged 41–80 years. Death occurred by hanging in 58.1% of these cases and poisoning in 12.4%. Suicide notes were most frequent in 2012 (22.9%) and least frequent in 2008 (5.7%); by month, they were most frequent in February (12.4%) and least frequent in September (4.8%). No significant difference was found in any study variable between those who left a note and those who did not except for the method used, with a note being recorded in 81.25% of poisoning cases (*p* = 0.040) (Table 1).

### 2.4. Data Analysis

IBM SPSS version 22 (IBM, Armonk, NY, USA) was used for statistical analyses. Quantitative variables were expressed as percentages and compared between cases with and without a suicide note using the chi-square (χ^2^) test.

For content analysis, primary documents were generated for each case, including sociodemographic data, forensic reports on the event and circumstances, and, when available, the text of suicide notes or observations on their presence/content. Primary documents were then reviewed and processed using ATLAS.ti version 7.5 (Scientific Software Development, Berlin, Germany). Two researchers (M.P.C.G., C.M.G.) undertook thematic coding of the texts, following the sequence described by Braun and Clarke [40,41]. Briefly, the texts were read to obtain an overview of the set and then to generate codes common to all cases divided between superordinate categories (themes) and subcategories. The researchers identified four main categories and 18 subcategories (Table 2), which were then confirmed by triangulation with two external researchers to establish a consensus. There were no discrepancies between researchers, and it was agreed that the topics were not exclusive and could coexist in the same text when necessary.

### 2.5. Ethical Aspects

The study complied with the principles of the Declaration of Helsinki and was approved by the research ethics committee of the Institute of Legal Medicine of Granada. Data were always anonymized and treated in accordance with national data protection legislation (Law 3/2018 of 5 December).

## 3. Results

### 3.1. Background and Context of the Act of Suicide

Table 3 displays forensic observations on the background of those dying by suicide. The most frequent reference was to the family context (21.32%), describing whether they lived alone or with others (parents, siblings, etc.), their marital/partner status (especially separation), the loss/disease of a close relative, a family history of suicide, and/or unspecified family problems. A history of at least one previous suicide attempt was reported in 11.53% of cases (23 males and 10 females). There were no reports of physical disease among the females versus seven reports among the males, and virtually the same number of males (14 cases) and females (13 cases) had a history of mental disorders, which were most frequently related to depressive states, followed by addictions (e.g., alcohol, drugs, or gambling).

Table 4 compiles observations on emotional states and recent events or situations that might be associated with the suicide and on the behaviors that immediately preceded it, including those related to the affective/family setting, health, and violence/abuse.

There were considerably more reports on males than females in all subcategories (61.18%). Descriptions of emotional states referred to explicit feelings (e.g., sad, discouraged, nervous, and/or depressed states, etc.), emotion-triggering situations (e.g., “his son committed suicide a few days ago”, “chronically ill and not cured”, etc.), and life events during the hours, days, or weeks before the death. These references were most often related to health status (20.27%), family (17.83%), and, to a lesser extent, economic (7.34%) and judicial events (3.49%). Health situations were related to worsening functional status and increased dependency, a recent diagnosis of cancer, Alzheimer’s disease, or schizophrenia, and the presence of unbearable pain or severe depressive symptoms. Recorded family events were mainly related to deaths, including highly traumatic cases (e.g., suicide of child or parent), and to marital/partner separation or family break-up. Economic situations included bankruptcy, eviction, and unemployment, while judicial situations included restraining orders, lawsuits, and trials.

Forensic observations of actions and behaviors in the hours before the suicide refer to different situations related to their family or health status. This information is based on evidence from third parties about telephone calls, family meals, leaving the children at school, and/or visits to health centers/emergency departments, including descriptions of the individual crying or watching family videos, among other related events. This type of report was more frequent in male than female cases.

Some reports on mistreatment and abuse referred to suicides by individuals after killing their partner, others described genital lesions in female cases, and reports from friends and family of previous intimate partner violence.

### 3.2. Suicide Notes

Of the suicide notes, 33.33% described in the forensic reports did not specify the addressee, 12.38% were expressly aimed at the family or a specific family member, and 22.85% addressed a specific person (e.g., by name or occupation). The note was described as handwritten in 18.09% of cases and as a mobile phone message in 1.90%, while this information was not given for the other notes mentioned. A single note was left in most cases, while in 17.24% of the cases, more than one note was left.

The contents of suicide notes were grouped into thematic codes (Table 5). Some of the notes shared various thematic codes. The majority were exclusively or in part farewell messages (39.04%), followed by notes that focused on explaining the reason for the suicide (26.66%) and those that left instructions on how to proceed or on the distribution of their goods or other requests (16.19%). There were also 17 notes (16.19%) seeking forgiveness or apologizing for what they were about to do. Interestingly, all of these were written by males except for one, in which a female expressed an apology but did not ask for forgiveness. Likewise, an explicit expression of affection was found in only one note from a female but in seven notes from males. Most notes were left by individuals aged between 41 and 60 years and featured messages of farewell, explanation, and/or apology.

## 4. Discussion

Forensic observations of the circumstances and contexts of death were available for around one-third of deaths by suicide in our setting. More than one-third of these observations referred to a suicide note (36.7%), although its textual content was only preserved in 17% of these cases. No significant differences in characteristics were found between those who left a note and those who did not, except for a higher frequency of poisoning among the former. Qualitative analysis of the forensic observations revealed the key importance of family context, previous suicide attempts, the diagnosis of a psychiatric disorder, and adverse life events, especially those related to health and family. Suicide notes were predominantly messages of farewell or explanation, and less frequently asked for forgiveness.

### 4.1. Forensic Observations

According to the forensic observations studied, the background of suicides most frequently involved family and relationship problems, health issues, especially psychological disorders, and a history of suicide attempts, in line with previous reports [42,43,44]. At the time of the suicide, the emotional state of the person was typified by sadness, anxiety, or failure, and there had been life-changing events or circumstances mainly related to loss of health, family breakdown, death of family members, including some by suicide, and, to a lesser extent, economic or judicial problems. In Spain, there are studies that suggest that the economic crisis can increase mortality from suicide and suicide attempts [45,46]. Nevertheless, in the study by Ruiz Ramos et al. about mortality in Andalusia, an increase in the mortality rate due to suicides in the economic crisis was not detected in either sex [47].

In addition, situations of gender violence, abuse, or mistreatment were recorded, as in previous studies [23,33]. The relationship between intimate partner violence and suicide attempts [48,49,50] and suicide [51] is well documented.

We highlight that some reported signs of impending suicide were missed, including visits to the family physician or hospital emergency departments. The significance of these and other behaviors was only grasped after the event, indicating failures in suicide risk assessment and follow-up care measures [42].

### 4.2. Suicide Notes

Suicide notes were left by a minority of these individuals, in line with previous studies worldwide [22,23,33,51,52]; however, unlike previous observations, there was a difference in suicide methods between those who left a note and those who did not, finding a significantly higher percentage of poisoning cases among the former. Rockett et al. [18] obtained a similar result but noted the difficulty of establishing intent in deaths by poisoning, except when a suicide note is present. Consequently, there may be an under-representation of suicides among deaths from poisoning and an overrepresentation of associated suicide notes.

In the present series, a narrative could be constructed from the notes that offered an explanation or apology for the act or sought forgiveness, revealing the role in their psychological distress of isolation, lack of affection, pain, disease, disability, and the desire to avoid being a burden to others. Previous studies have related suicidal behavior to functional disability, certain physical diseases, and mental disorders [53,54], and the resulting physical and emotional distress, loneliness, hopelessness [55], and pain [56]. However, one qualitative study reported that existential problems appear to be more important than mental disorders for individuals who choose to die by suicide [57].

More males than females died of suicide during the study period, and more males than females left a note, which may be related to differences between the sexes in the factors that underlie these acts [58]. Schrijvers et al. [59] reported gender differences in attitudes toward suicide and in the impact on suicidal behavior of mental or physical illness and psychosocial problems. The influence of physical or psychological gender violence on the mental health of women is well documented [49,60], and a study of forensic reports on suicides in the USA called for partner abuse to be evaluated as a risk factor [61].

The contents of the suicide notes were similar to those of previous findings [19,32,33,52,62,63,64]. It has been observed that the presence of a note indicates that the individuals remained connected to themselves and to others at the time of writing [22]. It is their last possible opportunity to explain or apologize for suicide and to offer comfort to their loved ones [33]. However, suicide notes are left in a minority of cases, and it remains controversial whether these are representative of suicide cases [19,22,65] or not [23,33,38,66]. In the present study, no difference was found between those who left a note and those who did not, except for the method of suicide.

It should be noted that the differences in the way in which the themes of the suicide notes are distributed in the studies consulted may be motivated by social and cultural aspects, or simply be an effect of the sample size, the way in which the information is collected, or even the sources of information. However, our study has highlighted results such as cases of gender violence, and the verification of the absence of notes in which women apologize for what they are going to do, or the minimum explicit expression of affection (only one case), compared to men. It is necessary to introduce the gender perspective in future studies.

### 4.3. Prevention Strategies

The traditional approach to prevention primarily considers the conditions in which suicides are most frequent, including some that are modifiable and others that are not. Greater focus on the person has been proposed, taking account of the emotional, psychological, and physical status of the individual. The observations and suicide notes examined in the present study offer insights into the life experiences and distress that lead to this act, helping to identify vulnerabilities that should be considered in the design of preventive strategies [27]. It has previously been observed that the content of a suicide note can provide crucial information about a person’s life situation to guide prevention [17,22,32]. In this way, steps could be taken to address the situations of loneliness that make some individuals vulnerable to suicidal thoughts. The diathesis–stress model of suicidal behavior developed over the past decade implicates the combination of an underlying vulnerability (diathesis) to suicide with adverse or stressful life events in deaths by suicide, and this new understanding of the phenomenon should be taken into account in the design of preventive measures [31,67,68,69].

We observed that 6.29% of the people who left a note before they died or had information from the coroner had gone previously, or a few days before, to the emergency services or to a specialist doctor for various problems, such as worsening of their condition, mental illness, anxiety, or suicidal thoughts. This situation shows that there are failures to detect and prevent impending suicides. Regardless of the reason why this problem was not assessed, either because it was not identified in the medical interview or the level of risk was not evaluated, it is important to consider it for suicide prevention training strategies for health professionals. In line with the person-centered approach to prevention, a stepped care model has been developed to assess suicide risk and to follow and treat vulnerable individuals at all levels of clinical care [70,71]. Likewise, in some autonomous communities of our country, early detection strategies such as the “Suicide Code”, a tool for the care and proactive monitoring of suicide attempts, are being implemented [72]. Another proposed continuity of care model focuses on young people, connecting those identified as most at risk with professionals trained in suicide prevention and ensuring that they continue to receive the mental health services they need [73]. In Andalusia, mental health care is mostly universal; there is a network of community and hospital care devices throughout its geography, carried out by teams of professionals from different disciplines, but it is necessary to establish a specific care strategy for people at risk of suicide. These approaches involve a structured continuum of activities across systems to facilitate coordinated care.

Finally, the authors emphasized the need for improvements in the quality of suicide risk assessment [28,42] and in the training of primary care physicians and nurses to detect vulnerable individuals at risk of suicide [74].

Through content analysis of forensic observations, complementary information has been obtained that helps us define epidemiological trends and stress factors in a specific geographic population. With this identification, we place ourselves in a more realistic and concrete position of the characteristics, with respect to the population at risk of carrying out suicidal-related behaviors, in our environment. Knowing these risk factors can help us to design suicide prevention strategies.

This retrospective study was limited to a specific time period and geographical area, and the results should therefore be interpreted with caution. The risk of information bias was minimized by preserving the textual notes written in the forensic autopsy report, with no interpretation by the researchers. Nevertheless, the reports were prepared by various professionals in different contexts who varied in their access to heterogeneous information and in their expectations, interests, and time availability. In this regard, it has been observed that the interpretation of ambiguous or complex scenarios by forensic physicians can be influenced by their specific training and experience [9,13,14].

## 5. Conclusions

This study of forensic observations confirmed the association of death by suicide with problematic life situations related to family and health and with emotional states of loss and sadness. Individuals who left a suicide note only differed from those who did not in the higher frequency of suicide by poisoning. The study of forensic observations and suicide notes improves understanding of the situations and states of mind that lead to this act and reveals failures by the health care system to detect and act on signs of vulnerability. This information is indispensable for supporting the design of well-targeted preventive strategies.

## Figures and Tables

**Table 1 ijerph-20-02281-t001:** Description of cases with and without a suicide note according to sociodemographic variables.

Variable		Total	Suicide Note		*p*
		(*n* = 286) *n* (%)	YES *n* = 105 (36.7%)	NO *n* = 181 (63.3%)	
Sex	Female	65 (22.7)	24 (22.9)	41 (22.7)	0.540 ^a^
	Man	221 (77.3)	81 (77.1)	140 (77.3)	
Age (years)	11–20	9 (3.1)	2 (1.9)	7 (3.9)	0.584 ^a^
	21–30	35 (12.2)	14 (13.3)	21 (11.6)	
	31–40	31 (10.8)	11 (10.5)	20 (11)	
	41–50	64 (22.4)	29 (27.6)	35 (19.3)	
	51–60	50 (17.5)	16 (15.2)	34 (18.8)	
	61–70	36 (12.6)	13 (12.4)	23 (12.7)	
	71–80	29 (10.1)	12 (11.4)	17 (9.4)	
	81–90	29 (10.1)	8 (7.6)	21 (11.6)	
	91 and over	3 (1)	-	3 (1.7)	
Method	Hanging	183 (64)	61 (58.1)	122 (67.4)	0.040 ^a^
	Knife	9 (3.1)	3 (2.9)	6 (3.3)	
	Firearm	10 (3.5)	4 (3.8)	6 (3.3)	
	Suffocation	5 (1.7)	3 (2.9)	2 (1.1)	
	Vehicular	4 (1.4)	1 (1)	3 (1.7)	
	Fire	3 (1)	1 (1)	2 (1.1)	
	C Poisoning	7 (2.4)	2 (1.9)	5 (2.8)	
	G Poisoning	6 (2.1)	4 (3.8)	2 (1.1)	
	D Poisoning	16 (5.6)	13 (12.4)	3 (1.7)	
	Fall	34 (11.9)	9 (8.6)	25 (13.8)	
	Drowning	7 (2.4)	3 (2.9)	4 (2.2)	
	Venesection	2 (0.7)	1 (1)	1 (0.6)	
Year	2007	24 (8.4)	13 (12.4)	11 (6.1)	0.157 ^a^
	2008	28 (9.8)	6 (5.7)	22 (12.2)	
	2009	42 (14.7)	16 (15.2)	26 (14.4)	
	2010	59 (20.6)	21 (20)	38 (21)	
	2011	40 (14)	13 (12.4)	27 (14.9)	
	2012	52 (18.2)	24 (22.9)	28 (15.5)	
	2013	41 (14.3)	12 (11.4)	29 (16)	
Month	January	26 (9.1)	12 (11.4)	14 (7.7)	0.741 ^a^
	February	26 (9.1)	13 (12.4)	13 (7.2)	
	March	28 (9.8)	11 (10.5)	17 (9.4)	
	April	22 (7.7)	8 (7.6)	14 (7.7)	
	May	32 (11.2)	8 (7.6)	24 (13.3)	
	June	21 (7.3)	7 (6.7)	14 (7.7)	
	July	28 (9.8)	9 (8.6)	19 (10.5)	
	August	23 (8)	8 (7.6)	15 (8.3)	
	September	16 (5.6)	5 (4.8)	11 (6.1)	
	October	30 (10.5)	10 (9.5)	20 (11)	
	November	15 (5.2)	8 (7.6)	7 (3.9)	
	December	19 (6.6)	6 (5.7)	13 (7.2)	

C = Caustic, G = Gas, D = Drug. ^a^ Chi-Squared.

**Table 2 ijerph-20-02281-t002:** Categories and subcategories in content analysis.

Category	Subcategories	Description
Sociodemographic data	Sex	Male, Female
	Age	Age at death: ≤10 years, 11–91 years; >91 years.
	Method	Hanging, Bladed Weapon, Firearm, Suffocation, Vehicular, Fire, Caustic Poisoning, Gas Poisoning, Drug Poisoning, Fall, Drowning, Venesection
	Suicide note: Yes/No	Yes: Presence recorded in forensic records. No: No reference to note in forensic records
Background	Previous suicide attempt	Previous suicide attempts reported to forensic institute
	Previous Physical Disease	Any physical disease or disorder
	Previous Psychiatric Disease	Any psychiatric disease or psychiatric/ psychological disorder.
	Previous family context	Reference in forensic report to family conditions/situations before or at the time of the suicide
Context of the act of suicide	Emotional State/Feelings	Emotional situation and feelings of the individual at the time of the suicide as described by third parties
	Recent family events	Family situation or family life events of the individual at the time of the suicide as described by third parties
	Recent health events	Health status or events at the time of the suicide
	Recent economic events	Economic situation or events related to the individual’s finances and employment at the time of the suicide
	Recent Judicial Events	Legal/judicial situation/event at the time of the suicide
	Previous affective/family events	Affective/family events or actions by the individual before the suicide.
	Previous medical/health acts	Health-related actions by the individual before the suicide
	Violence/Abuse	Situations or conditions of violence or abuse potentially related to the suicide
Suicide Note	Features	Medium (e.g., paper/text message, etc.), number of notes and addressee
	Content	Message of farewell, blame, explanation, affection, apology, instructions, and/or unspecified

**Table 3 ijerph-20-02281-t003:** Category: Background. Subcategories: number of cases, percentages, and verbatims of forensic observations in males and females.

Background Category	Number of Cases, %, and Verbatims		*N* = 286
	Male	Female	Total (%)
Previous suicide attempt	23	10	33 (11.53)
	Two attempts, by fall and venesection (C010, 70 years)	One previous attempt with medication 20 days earlier (C013, 63 years)	
	Three attempts (C185, 47 years)	Multiple attempts at self-harm aborted by husband (C060, 55 years)	
Previous Physical Disease	7	0	7 (2.44)
	End-stage cancer (C108, 50–59)		
	Amputated leg with orthopaedic prosthesis, awaiting surgery to amputate the other leg (C249, 85 years).		
	His wife says that he had been saying he wanted to be dead for some days because of his intense leg pains, which would not go away with any treatment (C266, 82 years)		
Previous Psychiatric Disease	14	13	27 (9.44)
	On anxiolytics since his wife’s worsening illness (C076, 73 years)	In treatment for depression (C004, 36 years)	
	...he was diagnosed with a depressive process (C139, 51 years)	... tired of suffering from bulimic disorder (C264, 35 years)	
Previous family context	46	15	61 (21.32)
	His father died in an accident 4 years ago which led him to take poison in a previous attempt (C033, 19 years)	Single, separated 3 months ago (C135, 37 years)	
	Separated with restraining order. Lived with his mother (C089, 51 years)	Suffering grief for the loss of his father (C026, 22 years)	
	Lived alone (C001, 43 years)	Lived alone (C051, 81 years)	

**Table 4 ijerph-20-02281-t004:** Category: Context of the act of suicide. Subcategories, number of cases, percentages, and verbatims of forensic observations in males and females.

Context of the Act of Suicide Category	Number of Cases, %, and Verbatims		*N* = 286
	Man	Female	Total (%)
Emotional State/Feelings	31	8	39 (13.63)
	He had been nervous for 15 days because his wife was operated on (C073, 89 years)	Thought his life was not worth living (C226, 51 years)	
	They had been talking the previous evening and found him sad and expressing words of farewell (C003, 82 years)	Alcoholic and had not been drinking for more than a year, but she had relapsed in the last two days (C269, 52 years)	
	Was partying the night before (C054, 22 years)	...she says she’s been acting strange for days (C123, 79 years)	
Recent family events	40	11	51 (17.83)
	Death of his wife a few months ago (C011, 83 years)	A few days before, she had just become a mother and could not breastfeed the baby (C004, 36 years)	
	In the process of separation (C121, 48 years)	His father also jumped to his death a few months ago (C093, 15 years)	
	Had threatened to do something after an argument with relatives (C030, 73 years)	Son committed suicide 15 days ago (C105, 51 years)	
Recent health events	42	16	58 (20.27)
	She was in a lot of pain from the carcinoma (C037, 68 years)	Psychiatric condition worsened two days before when his mother was admitted to a nursing home (C023, 60 years)	
	Recently diagnosed with Alzheimer’s (C124, 84 years)	Recent diagnosis of schizophrenia (C187, 37 years)	
Recent economic events	20	1	21 (7.34)
	Businessman, used to leave work at 4 am. The family says he had financial problems (C032, 53 years)	They had recently broken up and was also unemployed (C061, 36 years)	
	Seems that he would be expelled from the business that morning (C238, 53 years)		
Recent Judicial Events	7	3	10 (3.49)
	He had a restraining order against his wife for GV [Gender Violence] (C166, 57 years)	Was under litigation (C075, 62 years)	
	He had a judicial process on the same day (C189, 53 years)	Reported for gender violence (C163, 40 years)	
Previous affective/family events	17	6	23 (8.04)
	He called his wife saying he wanted to talk to his children for the last time (C086, 33 years)	Was watching the wedding video (C009, 27 years)	
	Had breakfast with colleagues, asked for a double cognac and sounded strange to the colleagues (C121, 48 years)	The whole family ate together, and he then went out to the barn and hanged himself during the siesta (C013, 63 years)	
	Been previously crying (runny nose) C054, 22 years)	Talked to the mother (C264, 35 years)	
Previous medical/health acts	12	6	18 (6.29)
	On the same day as the death, she went to her family doctor because of a worsening of her mental state (C100, 48 years)	She had been seen by the specialist the day before but had been discharged because there was no evidence of acute data (C019, 41 years)	
	Apparently, that night at about 4 o’clock he had gone to the emergency department of the Hospital... for anxiety (C183, 36 years)	He had gone to the family doctor that day. He had verbalized his intention to self-harm (C058, 62 years)	
Violence/Abuse	6	4	10 (3.49)
	The same day, the body of his girlfriend, with whom he lived, was found with signs of having been strangled hours before the suicide (C043, 34 years)	Thin. Recent anal tears and anal hematoma (C056, 22 years)	
	Two corpses are found in the c *, she is tied in supine decubitus position and he is in right lateral decubitus position, both naked. They are in the bedroom. (He has committed suicide by throat slashing) (C045, 50 years)	...says that she was physically abused in her past life but had never thought about suicide until now (C114, 47 years)	

* Initial corresponding to a specific location.

**Table 5 ijerph-20-02281-t005:** Category: Suicide notes: content of suicide notes, percentage, number of male and female cases and age ranges, and associated verbatims of forensic observations.

Suicide Note Contents	Total, %, Number of Cases and Verbatims											
	*N* = 105 Total (%)	Sex		Age								
		Man	Female	11–20	21–30	31–40	41–50	51–60	61–70	71–80	81–90	≥91
Affection	8 (7.6)	7	1	0	2	1	2	2	1	0	0	0
		He sent a message to his ex-wife by mobile phone saying that he was leaving this world and that he would always take her with him (C067) (male, 51–60 years) -“I am getting out of the way because without you I am nothing” (C219) (male, 41–50 years) -”[....] A kiss to all of you and remember me for my good things” (C142) (male, 61–70 years) - “Goodbye I love you all” (C239) (female, 51–60 years)										
Blame	5 (4.7)	4	1	0	0	1	1	2	0	1	0	0
		-He left a note saying that a family in the village is responsible for him taking his own life (C069) (male, 31–40 years). -Leaves farewell letter where she blames her husband for insulting her as “a drunk, pill-popper, whore, drug addict...” (C114) (female, 41–50 years) -He blames the Guardia Civil for the loss of points for traffic fines (C223) (male, 51–60 years).										
Farewell	41 (39)	33	8	1	6	3	15	7	4	4	1	0
		-Farewell for his children (C029) (male, 51–60 years) -Leaves some farewell messages on a T-shirt (C086) (male, 31–40 years) -Farewell note, saying that he didn’t know what was wrong with him, that he had something big inside him that wouldn’t let him live (C098) (male, 41–50 years) -She says goodbye to her son and siblings (C114) (female, 41–50 years) -Farewell note to his son. He says he couldn’t take any more pain (C148) (male, 51–60 years)										
Instructions	17 (16.1)	13	4	0	4	1	7	2	2	1	0	0
		-To be cremated and to take care of her children (C005) (female, 51–60 years) -Farewell message with instructions for the family (C015) (female, 41–50 years) -Has left his belongings (car, money) (C054) (male, 21–30 years) -He leaves a note in which he says he wants to be a donor and leave his body to science. He asks to give the news to his wife carefully and leaves her mobile number (C079) (male, 51–60 years) -“A *, everything on the table you take for yourself. Your father” (C112) (male, 71–80 years) -He distributes his belongings, phone numbers to let them know that he wants them to forgive him but he can’t do it anymore (C202) (male, 21–30 years)										
Justification	28 (26.6)	20	8	1	5	3	8	2	2	5	2	0
		-“If I have taken my own life, it is because I can’t stand the pain any more, it’s enough to make you mad” (C042) (male, 71–80 years) -She seems to have left a message saying she didn’t want to be a burden (C061) (female, 31–40 years) -She says that her husband was an ogre to her while he was nice to others. She says that what bothers her most is that she depends on what he chooses to give her (money for cigarettes, etc.) (C114) (female, 41–50 years) -...he wants to die, he commits suicide because he can no longer bear the loneliness, the lack of affection, his degenerative disease, he always has to sleep in cold-water towels, he wants to be reunited with his mother (C185) (male, 41–50 years) -“I’m very tired of suffering so much” (C190) (male 71–80 years) -“I’m fed up with life” (C221) (male, 11–20 years)										
Ask for forgiveness/ sorry	17 (16.1)	16	1	0	4	0	8	1	1	0	3	0
		-“Don’t blame anyone, I don’t want to suffer anymore, I ask for forgiveness from my wife, children, family and friends” (C092) (male, 81–90 years) -“I ask you all for forgiveness, my children, grandchildren, parents, siblings, you, my wife... so many years together. Goodbye to all” (C101) (male, 41–50 years) -“I know it’s cowardly, but I can’t go on any longer. I hope you will forgive me. A kiss to all of you and remember me for my good things” (C142) (male, 61–70 years) -“I write these letters to ask for forgiveness from all the people who feel affected by my decision, I do it on my own free will and convinced that I have no other way out and perhaps no desire to look for it. [...] I am sorry from the bottom of my heart that it has come to this” (C156) (male, 41–50 years) -“I’m sorry for the trouble I’m going to put you through. I wish I could disappear from the world and memory, but I can’t. Some of us are meant to be unhappy forever. It’s incurable. Thank you for everything. Goodbye” (C206) (male, 21–30 years) -“I’m sorry I didn’t pay you what I owed you. Now you can take advantage of me if you want. But excuse me, I’m even colder” (C056) (female, 21–30 years)										
Not specified	16 (15.2)	11	5	0	2	1	2	2	5	2	2	0
		-Leaves note with intent to commit suicide (C059) (female, 61–70 years) -Suicide note exists (C137) (male, 41–50 years)										

* Initial corresponding to omitted name. Quotation marks are used when the words are verbatim.

## Data Availability

Data not available due to ethical and legal restrictions.

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
