# Peer review of "Understanding for Prevention: Qualitative and Quantitative Analyses of Suicide Notes and Forensic Reports"

_ijerph, 2023, doi:10.3390/ijerph20032281_

Round 1

Reviewer 1 Report

The current manuscript describes a quantitative and qualitative examination of forensic reports and suicide notes associated with suicide deaths in Spain. Suggestions for revision are below by manuscript section instead of in order of importance.

Introduction:

1. Given that the age range in the present study is 11-91 years, the introduction would be better served by providing broad rates of global suicide death instead of honing in on 15-29 year olds. This sets the reader up to believe the focus of the present study will be on that specific age group.

2.  In lines 42-43  (p. 1), the authors note that the forensic physician obtains information on motives, characteristics, etc. Can the authors provide more detail on how this is approached, given that this is an aspect that feeds directly into the forensic reports they are analyzing?

3. Lines 61-70 (p. 2) feel somewhat out of place with their description of the importance of person-centered approaches. Given that a large focus of this analysis is on forensic reports which are, inherently, not person-centered as they are completed by an individual not known to the decedent. Suicide notes can provide some person-specific and person-centered information, but I'm unclear on how third party forensic reports can achieve this. Can the authors clarify?

Materials and Methods:

1. What data were missing from the other 457 suicide deaths that they could not be included in the analyses? Were these individuals different from the included decedents in any way (e.g., age, sex)?

2. Can the authors report a M/SD of overall age?

3. Given that a large focus of this investigation is qualitative, a significant amount of methodological detail is missing. Can the authors provide additional detail on their approach in-text? For example: How were codes generated? Were codes based on prior work? How did authors ensure reliability? How were discrepancies resolved?

Results:

1. For Table 1, please report your test statistic in addition to the p-value. Please add a note to the table about what statistical test(s) was performed to examine group differences.

2. For the categories in Table 3, how did the authors address if any of the decedents fell into multiple categories? For example, a decedent could have had both a previous suicide attempt and a previous physical disease.

3. Please report percentage of decedents with a prior history of suicide attempt in Table 1 if the data is available.

4. Line 77 (p. 4): 33% is not most of your sample of suicide notes.

5. In the suicide notes results section, please provide the number of suicide notes. How many decedents left more than one suicide note?

Discussion:

1. The authors review their two streams of data (i.e., forensic report, suicide note) separately, but there is little discussion on how we might integrate these streams of data to better understand suicide from an 'objective' (forensic report) and 'subjective' (suicide report) point of view. What might we learn from integration? What challenges might there be in integrating these two data streams?

2. Line 240 (p. 5): please use "died by suicide" instead of "committed suicide." Please also check the rest of the manuscript for this language use. 

3. Regarding the contents of the suicide notes (p. 6), did the authors find anything that was unique or different than what has appeared in current literature?

4. Lines 270-272 (p. 6): how did the authors determine that suicide wasn't asked about in general practitioner or hospital visits? If the authors do not have data to support this, they are encouraged to broaden their statement and note that medical appointments may be an important place for suicide screening and they can discuss the importance of suicide assessment training.

Author Response

The current manuscript describes a quantitative and qualitative examination of forensic reports and suicide notes associated with suicide deaths in Spain. Suggestions for revision are below by manuscript section instead of in order of importance.

Dear Reviewer,

First, we want to deeply thank your thoughtful review that, in our opinion, has improved and clarified our manuscript.

You can find below (in italics) the responses to the comments and suggestions. The text added or modified in the manuscript is highlighted in red (also in the manuscript file).

Yours cordially,

The authors

Introduction:

  1. Given that the age range in the present study is 11-91 years, the introduction would be better served by providing broad rates of global suicide death instead of honing in on 15-29 year olds. This sets the reader up to believe the focus of the present study will be on that specific age group.

Regarding this suggestion, we have changed the sentence and completed some data in lines 28-30, as follows:

“In 2019, 1.3% of deaths were by suicide and the global age-standardized suicide rate was 9.0 per 100,000 inhabitants (2.3 times higher in men than in women). This potentially preventable act occurred before the age of 50 in over half of global suicides [3]

  1. In lines 42-43  (p. 1), the authors note that the forensic physician obtains information on motives, characteristics, etc. Can the authors provide more detail on how this is approached, given that this is an aspect that feeds directly into the forensic reports they are analyzing?

Thank you for your suggestion. According to it, we have added some more information to clarify this:

“Therefore, the information is obtained in the first instance by direct observation of the scene of the events and by questioning the people present (relatives, neighbours, acquaintances). Later, the information obtained in the clinical autopsy process (concomitant causes, substance use, time of death, etc.) is added. This direct approach allows to observe where the person chose to die (in what area of the home, in the mountains, workplace, etc.) and in what way (method), how they prepared the scene (if it was an impulsive act or, on the contrary, it was carefully prepared, leaving even the necessary documentation near the body) and if the deceased left any note before he died”. (Lines 50-58)

  1. Lines 61-70 (p. 2) feel somewhat out of place with their description of the importance of person-centered approaches. Given that a large focus of this analysis is on forensic reports which are, inherently, not person-centered as they are completed by an individual not known to the decedent. Suicide notes can provide some person-specific and person-centered information, but I'm unclear on how third party forensic reports can achieve this. Can the authors clarify?

Data collection from the scene and transcription of the notes, is conducted by a professional that “takes a photo” of the staging that the person that died by suicide, on an intimate way, performed before dying, in order to clarify some specific aspects that determined the suicidal behaviour.

In our public health system, specific models of attention to people in risk of suicide are underdeveloped, and Mental Health professionals do not have tools to prevent or measure objectively and effectively the risk of a person committing a suicidal act.

Furthermore, there is no record that all the people who died by suicide were included in the Mental Health system. Therefore, from our personalized care approach, our attention is focused on the identification of individual risk factors, which leads us to a person-centered assessment, hence, the importance of continuing to understand this behaviour individually when the person has died by suicide. Forensic observations can help to know and relate factors that conditioned this behaviour. In this way, we ensure a more person-centred approach, based on the highly personal and fluid nature of suicide risk, which is related to individual vulnerabilities and person-specific events that trigger suicidal thoughts and actions [27, 30, 31].

A new paragraph has been added to clarify this idea:

Thus, forensic reports can offer personalized information when they include direct observations and information collected from family and friends about the conditions and circumstances of death. These person-centered observations may help and/or supplement the absence of suicide notes or psychological autopsy reports.” (Lines 85-89)

Materials and Methods:

  1. What data were missing from the other 457 suicide deaths that they could not be included in the analyses? Were these individuals different from the included decedents in any way (e.g., age, sex)?

Thank you very much for your comment. We excluded those cases in which the revised forensic report did not contain any different or complementary comments on the conditions and circumstances of the deceased (and of suicides) that should be included in the mandatory clinical autopsy. There were no individual differences, but differences in the data included by the coroner in his/her report on each case.

To clarify this issue, we included the following sentence in lines 119-120:

“… (complementary information to the mandatory clinical autopsy data)”

  1. Can the authors report a M/SD of overall age?

Sure, thank you very much for your suggestion. We have included the mean/SD age in the text:

“Mean age was 52.99 years (±19.135) and the individual…” (Line 120)

  1. Given that a large focus of this investigation is qualitative, a significant amount of methodological detail is missing. Can the authors provide additional detail on their approach in-text? For example: How were codes generated? Were codes based on prior work? How did authors ensure reliability? How were discrepancies resolved?

Thank you for your thorough revision.

Codes were generated from the text of the forensic observations included in the forensic reports. They were a very descriptive text, without explanations or different interpretations of the coroner’s impressions or someone’s direct quotes (e.g. relatives…). The forensic observations limited to explain the situation as it was. This can be observed on the direct quotes or verbatims, of the analyzed cases, contained in tables 4 and 5.

Codes were grouped in categories and subcategories according to the main themes and subthemes. Table 2 specifies this information with the individual description.

For reliability, the coding carried out by the two researchers, who initially agreed on this structure of topics and subtopics for analysis with the Altas ti software, was reviewed, after a detailed reading of all the cases analyzed. This review was carried out by triangulation with two external investigators. There were no discrepancies on this thematic structure, so no modification was made. This was mainly because an agreement was reached that in the same text or verbatim, several themes could coexist and therefore they were not exclusive.

To clarify this in the text, we have included the following sentence in lines 149-150:

“…There were no discrepancies between researchers and it was agreed that the topics were not exclusive and could coexist in the same text when necessary.”

Results:

  1. For Table 1, please report your test statistic in addition to the p-value. Please add a note to the table about what statistical test(s) was performed to examine group differences.

Thank you very much for your suggestion. The statistical used was Chi-Squared, we added this information in Table 1.

  1. For the categories in Table 3, how did the authors address if any of the decedents fell into multiple categories? For example, a decedent could have had both a previous suicide attempt and a previous physical disease.

Indeed, as explained, this could happen, and they were counted in all the categories/subcategories affected.

  1. Please report percentage of decedents with a prior history of suicide attempt in Table 1 if the data is available.

This percentage is specified in Table 3.

According to data from the forensic registries, a 11.53% (23 men and 10 women) of the 286 cases presented previous suicide attempts.

  1. Line 77 (p. 4): 33% is not most of your sample of suicide notes.

Thank you very much for this suggestion. We have corrected the expression as follows:

“The 33.33% of the suicide notes…” (Line 194)

  1. In the suicide notes results section, please provide the number of suicide notes. How many decedents left more than one suicide note?

Thank you for this comment. 105 cases left a suicide note and 18 left more than one note (17.14%). We have added this in the text, in order to clarify:

“…while on 17.24% of the cases, more than one note was left.” (Line 199)

Discussion:

  1. The authors review their two streams of data (i.e., forensic report, suicide note) separately, but there is little discussion on how we might integrate these streams of data to better understand suicide from an 'objective' (forensic report) and 'subjective' (suicide report) point of view. What might we learn from integration? What challenges might there be in integrating these two data streams?

We understand your interest in an integrative approach. Without a doubt, it really is a very interesting approach, however, with the available data, this is not possible. Actually, there are not two data streams. All data comes from observations of forensic reports, including data from suicide notes. It was not possible to access the original suicide notes, so what we know is what the forensics included in their reports about them. That is why it has been treated as a category of content analysis, and not as a differential analysis. This is mentioned in the limitations of the study.

For this reason, we also understand that an important contribution of this work is that it values the forensic information about the circumstances and contexts of the death to get closer to a knowledge of the act, what precedes it and what surrounds it.

In this study about the population that died by suicide on a 7 year period in a province, we have analyzed data reported by forensic reports as: sociodemographic characteristics, medical, personal and family data and those aspects related to the suicidal act as temporality, method and the existence of a suicide note left by the person died by suicide.

Furthermore, through the content analysis of the notes, we wanted to know what had motivated this final decision of the person died by suicide. This complementary information help us define new epidemiological trends and stressful factors on a concrete geographical population. This identification, places ourselves in a more realistic and concrete position of the characteristics, with respect to the population at risk of carrying out suicidal behaviour, in our environment. Knowing these risk factors can help us design suicide prevention strategies and also to compare with other populations from different geographical areas.

To reinforce this idea, the following paragraph has been added in the manuscript, at the end of the discussion:

“Through content analysis of forensic observations, complementary information has been obtained that helps us to define epidemiological trends and stress factors in a specific geographic population. With this identification, we place ourselves in a more realistic and concrete position of the characteristics, with respect to the population at risk of carrying out suicidal-related behaviours, in our environment. Being able to know these risk factors can help us to design suicide prevention strategies.” (Lines 318-323)   

  1. Line 240 (p. 5): please use "died by suicide" instead of "committed suicide." Please also check the rest of the manuscript for this language use.

Thank you for this appreciation. Even though we used the term “died by suicide”, seems that “commited suicide” appeared as an error. We have corrected it in (actual) line 267. Other sentences that include “commited suicide” instead of “dead by suicide” are original direct quotes or verbatims from the forensic reports that cannot be modified.

  1. Regarding the contents of the suicide notes (p. 6), did the authors find anything that was unique or different than what has appeared in current literature?

According to the literature, something curious happens in the content of suicide notes. It seems that the themes are universal; however, they differ in the way they occur in different places.

In our study, it also happens that while there are common themes, they differ in some aspects of their presentation.

For example, compared with the study by Ceballos-Espinoza (Ceballos-Espinoza, F. (2014). Suicidal speech: An approach to the meaning and meaning of suicide based on the analysis of suicide notes. Gaceta de Psiquiatría Universitaria, 10(3 ), 350-357) in which the same categories of our study appeared, these contents differ in the percentage of appearance (higher in the content of requesting forgiveness and/or apologies for the suicidal act, and more frequent in women than in men in our study). These differences may be motivated by social and cultural aspects, or simply be an effect of the sample size or the way in which the information is collected, or even the sources of information.

In this sense, ignoring these small differences with other national or international studies, we consider that there are two original or newer contributions to the subject: we would like to highlight the cases found for gender violence, a scourge of today's society, in which women not only die murdered, but it is also a cause of death by suicide, as we have been able to show in 4 cases out of the 286 cases included in our study; and also the verification that no woman asked for forgiveness (only one apologized for what she was going to do) and only one case showed an explicit expression of affection.

Probably, the fact that many fewer women than men die by suicide has been a handicap for delving into the underlying reasons for suicide in women. It is clear that deaths by suicide can be very different in men and women, so it seems necessary to introduce the gender perspective in the analyzes carried out.

This leads us to consider, in a special way, the situation of women who died from this act in assessing the risk of suicide.

We tried to complete the discussion, adding a paragraph at the end of the “Suicide Notes section”:

“It should be noted that the differences in the way in which the themes of the suicide notes are distributed in the studies consulted may be motivated by social and cultural aspects, or simply be an effect of the sample size, the way in which the information is collected or even the sources of information. However, our study has highlighted results such as cases of gender violence, and the verification of the absence of notes in which women apologize for what they are going to do, or the minimum explicit expression of affection (only one case), compared to men. It is necessary to introduce the gender perspective in future studies.” (Lines 272-279)

  1. Lines 270-272 (p. 6): how did the authors determine that suicide wasn't asked about in general practitioner or hospital visits? If the authors do not have data to support this, they are encouraged to broaden their statement and note that medical appointments may be an important place for suicide screening and they can discuss the importance of suicide assessment training.

Thank you very much for your revision, it has clearly improved the manuscript.

According to this suggestion, we have include/modified the following paragraph in the discussion section:

“We have observed that 6.29% of the people who left a note before they died or had information from the coroner had gone previously, or a few days before, to the emergency services or to a specialist doctor for various problems such as worsening of their condition, mental illness, anxiety, or suicidal thoughts. This situation shows that there are failures in the detection and prevention of impending suicides. Regardless of the reason why this problem was not assessed, either because it was not identified in the medical interview or the level of risk was not evaluated, it is important to consider it in suicide prevention training strategies for health professionals. Following the person-centered approach to preventing suicide, a stepped care model has been developed to assess suicide risk and to follow and treat vulnerable individuals at all levels of clinical care [74,75]. Likewise, in some autonomous communities of our country, early detection strategies such as the "Suicide Code", a tool for the care and proactive monitoring of suicide attempts, are being implemented [76*].” (Lines 304-306)

*New reference added.

Reviewer 2 Report

The manuscript describes a mixed-methods study of forensic report and suicide letters from Spain. Suicide prevention remains an important field of action and basically the paper is written in a very clear and readable way. From my point of view, however, the gain in knowledge is very limited. I list the most important reasons for this here: 

1. the information is highly selective, because forensic reports are available only in part, and for even fewer persons also suicide notes. The information from the suicide notes offered little information for suicide prevention. 

2. the findings are ultimately not new and, by and large, this information has already been corroborated in  methodologically sound studies, for example, prospective studies. Therefore, unfortunately, the question is what we learn here that is new and robust. 

3 If we really want to improve suicide prevention, we need studies that examine the effectiveness of interventions. These already exist, although there are still many questions. In my view, a highly selective and retrospective assessment of the circumstances of a suicide will not help us. 

For the above reasons, I cannot recommend the manuscript for publication. I regret this very much, because it is basically well readable and clearly written and certainly a lot of work lies in it. However, if you look at the findings on the prediction of suicides (especially here meta-analysis by Franklin, 2017), it becomes clear that we have a fundamental problem with the prediction of suicides and the research is nevertheless already much further than what is presented here in the paper. 

Author Response

The manuscript describes a mixed-methods study of forensic report and suicide letters from Spain. Suicide prevention remains an important field of action and basically the paper is written in a very clear and readable way. From my point of view, however, the gain in knowledge is very limited. I list the most important reasons for this here:

Dear Reviewer,

Thank you for taking the time to read and analyse our manuscript.

Please, find below our responses to your suggestions and comments.  We highlighted them in italics.

Yours cordially,

The authors

  1. the information is highly selective, because forensic reports are available only in part, and for even fewer persons also suicide notes. The information from the suicide notes offered little information for suicide prevention.

Indeed, that is the situation of forensic information. In Spain, to access forensic reports from the autopsy, many permits and special authorizations are needed, since the notes of people who die by suicide are considered judicial evidence.

On the other hand, we find that a contribution of this study is, precisely, to show the importance of a good and complete collection of information when the act occurs. And it is important to do this, in order to understand and have a better real evidence of the conditions and context of the people died by suicide. Forensics can become aware of the relevance of their work beyond the specifically clinical autopsy, obviously, of enormous importance.

  1. the findings are ultimately not new and, by and large, this information has already been corroborated in methodologically sound studies, for example, prospective studies. Therefore, unfortunately, the question is what we learn here that is new and robust.

This is a study on the general population, which most innovative and solid contributions could be summarized as follows:

  • It has been more than demonstrated that a generic prevention of deeply personal and individual behavior produces very limited results, only feasible for a specific social and cultural environment, and therefore has little impact on people who may be considering carrying out this act. Therefore, it is necessary to know the specific population that may be at risk in a community, which may, indeed, share characteristics and contexts with others, but we do not know this until we study it.
  • The scarcity of suicide notes is itself, a very important fact to understand that whoever performs this act does not want to, cannot, does not know or has nothing to say before dying. It can be a clear proof of the utter loneliness and isolation of that person, or of the impulsiveness of his act, or perhaps it does not mean anything. To corroborate it once more is to add evidence to a situation that seems universal, but that does not mean that it is identical in all places. We show that there are no verifiable differences with the variables analyzed in our study between those who leave a note and those who do not. Therefore, we can consider that the situations of both may be similar.
  • We can highlight specific contributions related to the gender perspective, as we have indicated to reviewer 1. We have included a new paragraph in the manuscript to clarify this idea:

“Through content analysis of forensic observations, complementary information has been obtained that helps us to define epidemiological trends and stress factors in a specific geographic population. With this identification, we place ourselves in a more realistic and concrete position of the characteristics, with respect to the population at risk of carrying out suicidal-related behaviours, in our environment. Being able to know these risk factors can help us to design suicide prevention strategies.” (Lines 318-323)

3 If we really want to improve suicide prevention, we need studies that examine the effectiveness of interventions. These already exist, although there are still many questions. In my view, a highly selective and retrospective assessment of the circumstances of a suicide will not help us.

As the reviewer expresses, it is a personal opinion. In our case, we think that knowing the specific characteristics of the population in our environment helps us to carry out prevention strategies adapted to the evidence found.

For the above reasons, I cannot recommend the manuscript for publication. I regret this very much, because it is basically well readable and clearly written and certainly a lot of work lies in it. However, if you look at the findings on the prediction of suicides (especially here meta-analysis by Franklin, 2017), it becomes clear that we have a fundamental problem with the prediction of suicides and the research is nevertheless already much further than what is presented here in the paper.

We share your concern about the problem of predicting death by suicide, even if the approach differs. There is, certainly, not just one way to do it. Although much progress has obviously been made in suicide prevention research, reality shows us that deaths by suicide are increasing.

Referring to the meta-analysis by Franklin et al. (2017), we did not find in it information regarding suicide notes or psychological autopsy. In our opinion, these sources of information can be a fundamental pillar in the understanding of this phenomenon; that although it requires the identification of risk factors and statistical analyzes that can help predict the risk of suicide through algorithms, cannot explain the complexity of the human being and its way of interpreting life. That is why our study is committed to understanding the complexity of this phenomenon.
